# Bioelectromagnetism for Cancer Treatment—Modulated Electro-Hyperthermia

**DOI:** 10.3390/curroncol32030158

**Published:** 2025-03-11

**Authors:** Andras Szasz

**Affiliations:** Department of Biotechnics, Hungarian University of Agriculture and Life Sciences, 2100 Gödöllő, Hungary; biotech@gek.szie.hu

**Keywords:** thermal, nonthermal, resonance, modulation, electro-hyperthermia, oncotherapy

## Abstract

Bioelectromagnetism has the potential to revolutionize cancer treatment by providing a noninvasive, targeted, and potentially more effective complement to traditional therapies. Among bioelectromagnetic techniques, modulated electro-hyperthermia (mEHT) stands out due to its unique characteristics, which have been supported by experimental evidence and clinical validation. Unlike conventional hyperthermia methods, mEHT leverages nonthermal bioelectromagnetic processes, offering a distinct and promising approach in oncology. This differentiation underscores the broader potential for bioelectromagnetic applications in cancer treatment, paving the way for innovative therapeutic strategies.

## 1. Introduction

The potential of bioelectromagnetism to address the therapeutic gap and resistance in current oncological treatments is inspiring. Current oncological treatments often reveal a disparity between the existing state of cancer care and the ideal or optimal standards that could be achieved. Although many cancer treatments can be highly effective, they frequently come with significant side effects. There is a notable gap in managing these side effects and enhancing the overall quality of life for patients undergoing treatment. This gap arises because the growth of cancerous masses tends to be exponential, while the cell distortion associated with malignancy is typically linear or quadratic, as described in radiotherapy [1]. Efforts to compensate for the rapid tumor growth with increased doses are limited by the toxicity of the methods, resulting in a therapeutic gap (Figure 1a). Additionally, cancer cells can develop resistance to therapies over time, further reducing the effectiveness of current treatments and exacerbating the therapeutic gap (Figure 1b). Addressing and overcoming this resistance remains a significant challenge in oncology.

The therapeutic gap manifests in several ways:•Access Disparities: Not all patients have equal access to the latest and most effective therapies. Factors such as a patient’s personal health, diet, geographic location, socioeconomic status, and healthcare infrastructure can significantly impact their ability to receive cutting-edge treatments.•Customization Limitations: Although advances in personalized medicine have enabled tailored treatments based on individual genetic profiles, there are still limitations in the extent to which therapies can be customized. This gap may lead to less effective treatments for some individuals.•Stage of Diagnosis: The effectiveness of oncological treatments often depends on the stage at which cancer is diagnosed. Gaps in early detection methods can result in diagnoses at more advanced stages when treatment options may be less effective.•Mortality Rates: Despite new treatments, the mortality rate for some cancers does not decrease significantly. Factors such as an aging population, environmental exposures, and socioeconomic disparities contribute to this persistent issue. Limited access to care, high treatment costs, and geographic disparities further exacerbate mortality rates. Effective therapies are still lacking for many cancers, highlighting the need for ongoing research to develop new treatments and improve our understanding of cancer mechanisms.•Integration of Emerging Technologies: Emerging technologies such as immunotherapies, targeted therapies, and precision medicine are not always swiftly integrated into standard practice. Bridging this gap involves translating research breakthroughs into clinical practice efficiently and effectively.

Addressing the therapeutic gap necessitates the development of innovative techniques that are minimally toxic, adaptable to the treated subject, complementary to existing therapies, and cost-effective. Ideally, these new technologies should be able to modify the chemical reactions within living organisms without using drugs. Bioelectromagnetism represents a promising candidate for this role, provided its targeting and processes are carefully controlled. Our current objective is to study and explore this potential.

## 2. Electromagnetic Characteristics of Malignant Cells

Cancer cells often exhibit an altered resting membrane potential compared to healthy cells [2]. Cancer cells typically depolarize this potential, influencing various cellular processes, including growth and proliferation [2]. This alteration is partly driven by the Warburg effect, which describes how metabolic reprogramming provides cancer cells with several advantages for growth and survival. The Warburg effect is a metabolic hallmark of cancer cells characterized by increased glucose uptake, lactic acid production, and reduced mitochondrial activity; the cancer cells preferentially undergo aerobic glycolysis, producing lactate even in the presence of oxygen. The resulting increase in extracellular acidity can modify the electrical properties of tissues by altering the ionization state of various molecules and influencing cellular interactions.

Ion channel activity in cancer cells also differs from that in normal cells [3]. The acidic environment resulting from the Warburg effect [4] can affect the function of ion channels and transporters on the cell membrane, contributing to the altered membrane potential. This metabolic shift influences ion channel activity, leading to altered expression of channels that maintain ionic balance, which is crucial for cancer cell survival and proliferation. These changes can impact electrical properties such as membrane resistance and capacitance, with upregulated or mutated ion channels leading to increased ionic currents and altered electrical activity.

Cancer cells and tissues often display different electrical impedances in comparison to healthy counterparts [5]. The accumulation of lactate and other metabolites can influence the conductivity of cancer cells, as acidic conditions may alter the dielectric properties of cells and the surrounding extracellular matrix (ECM), resulting in lower impedance in cancer cells due to differences in cellular structure and density. These changes affect how electrical currents and electromagnetic fields interact with the tissue. The lower impedance facilitates the flow of electric fields and currents through the cancer tissues [6].

The shift to glycolysis in cancer cells also involves changes in the levels and activities of various glycolytic enzymes [7], which can impact the cells’ bioelectromagnetic properties. These enzymes may influence intracellular ionic concentrations and electrical signaling pathways [8]. Increased glycolysis generates more metabolic heat due to higher glucose turnover [9], potentially affecting the thermal properties of cancer cells and their interaction with electromagnetic fields, thus influencing the efficacy of electromagnetic therapies.

The ECM in tumors often significantly differs from that in normal tissues [10]. The Warburg effect contributes to changes in the tumor microenvironment, including ECM alterations, which can impact tumor tissue’s electrical impedance and conductivity. ECM remodeling in tumors can modify how electrical currents are distributed and how electromagnetic signals are absorbed or scattered, with changes in the composition and structure of the ECM affecting its conductivity and the transmission of electrical signals through the tissue [11].

Cells emit various electromagnetic signals due to their intrinsic electrical and biochemical activities [12,13], and this inherent noise is characteristic of complex systems. These signals can be categorized based on different aspects of cellular function and their interaction with electromagnetic fields. Due to their deviated metabolism, cancer cells, particularly, can emit distinct bioelectric signals compared to normal cells [14].

Complex, self-organized, self-similar structures characterize the intrinsic organization of living organisms [15]. This self-organized complexity is reflected in the emitted spectrum of chemical activity in cells, manifesting as characteristic fractal dynamics over time [16]. The time–fractal fluctuation of homeostatic processes is associated with noise, where the logarithm of the energy spectral density inversely depends on the frequency of the emitted noise (1/f noise), a characteristic of the homeostatic operation of tissue [17].

## 3. Modulated Electro-Hyperthermia (mEHT)

### 3.1. Mechanisms of mEHT

The key principle of mEHT focuses on the bioelectromagnetic and thermal differences between healthy and malignant cells [18]. Cancer cells exhibit several distinct bioelectromagnetic properties compared to their healthy counterparts. By harnessing well-designed bioelectromagnetic interactions, mEHT can selectively target malignant cells, acting heterogeneously rather than uniformly [18], and effectively complementing conventional oncotherapies. These bioelectromagnetic differences are primarily due to the altered cellular and molecular environment in cancerous tissues, allowing for precise targeting of malignancies.

A critical factor contributing to cancer cells’ divergent behavior is their heightened metabolic activity, which fosters their autonomy and disrupts the healthy cellular network. Malignant cells often prefer aerobic glycolysis over oxidative phosphorylation for energy production. This metabolic irregularity is closely linked to cancer cells’ electromagnetic properties. The mEHT uses these irregularities to select these cells and destroy them with a synergy of thermal and nonthermal energy from the radiofrequency (RF) electric current [19].

### 3.2. Technical Implementation of mEHT

The technical realization of mEHT involves a highly precise impedance-matched resonant circuit [20] that adapts to real-time changes in the target tissue. Energy delivery is managed through a dominant electric field in the near-field arrangement of a capacitive coupling, using accurately and dynamically corrected impedance matching [21]. This sensitive tuning technique maximizes treatment efficacy, forming a galvanic-like connection between the electrodes and the patient’s body. The patient becomes an integral part of the resonant circuit (Figure 2).

Living systems are regulated by a complex dynamic equilibrium, adapting to environmental challenges [22], including those posed by treatments. mEHT employs a stochastic approach [23], acknowledging the time-dependent nature of biological processes, rather than relying on a deterministic framework [24] that overlooks homeostatic dynamism [25].

### 3.3. Electromagnetic Frequency Dispersions and mEHT

In general, the effects of RF radiation on living systems depend on the frequency and intensity of the radiation, as well as the type of tissue that is exposed. Tissues exhibit three primary electromagnetic frequency dispersions (α, β, γ) [26], corresponding to relaxational processes. One way of relaxation is through dielectric polarization, which occurs when the electric field of the RF radiation causes the molecules to become polarized. The molecules then try to align themselves with the electric field, which can cause them to rotate and vibrate. The relaxation time refers to the time it takes for a system to return to equilibrium after being disturbed. It can be as follows:•Molecular relaxation, which takes molecules to return to their original energy state after absorbing radiation;•Stress relaxation, which is when the mechanical or electric stress gradually decreases in the tissue under constant strain;•A chemical reaction to reach equilibrium after a perturbation.

The relaxation processes differ by frequency ranges, usually defined at low frequencies (α), radio frequencies (β), and microwave frequencies (γ) [27]. The preferred frequency for mEHT, based on international standards for industrial, scientific, and medical use, is 13.56 MHz. In silico models have demonstrated the importance of this frequency for mEHT [28].

However, while the 13.56 MHz frequency effectively selects cancer cells, the nonthermal polarization changes necessary for molecular restructuring and intracellular signaling require low frequencies in the α-dispersion range [29,30,31]. To address this challenge, mEHT uses a 13.56 MHz carrier signal with low-frequency amplitude modulation from the α-dispersion range. Impedance matching optimizes thermal and nonthermal energy transfer, and the precise tuning process also offers diagnostic value [32,33], making mEHT a theranostic technique. The application of 1/f modulation further helps regulate biosystems toward standard homeostasis [34], enhancing the selective destruction of malignant cells.

### 3.4. Thermal and Nonthermal Processes in mEHT

mEHT leverages the bioelectromagnetic differences between cancerous and healthy cells to achieve targeted treatment. This technique focuses on the unique electrical properties of cancer cells, utilizing bioelectromagnetic interactions to disrupt malignant processes while cooperating with healthy homeostatic mechanisms [35]. These bioelectromagnetic interactions significantly impact the chemistry of malignant tissues, affecting cellular processes, metabolic pathways, and biochemical environments, and could cause cell death. They also influence ion channels, the extracellular matrix (ECM), and protein functions, which can alter tumor growth, response to therapy, and overall cellular behavior.

Electromagnetic fields, particularly those in the radiofrequency range, can cause localized heating of tissues. This thermal effect can modify biochemical reactions by changing tissue temperatures, potentially impacting enzyme activity and metabolic rates. mEHT represents a unique advancement in oncological hyperthermia by combining nonionizing electric fields with thermal effects to target cancer cells more precisely. Unlike conventional hyperthermia, which focuses solely on thermal effects, mEHT integrates thermal and nonthermal processes to enhance treatment efficacy, selecting and destroying the malignancy.

Traditional hyperthermia approaches, rooted in Hippocratic principles, focus on isothermal heating, aiming to evenly distribute temperature across the tumor. However, this approach is flawed because heat disperses from the targeted area, and thermal homeostasis increases blood flow, which may elevate the risk of metastases [36]. The complexity of deep-focused heating reflects the intricate physiological and technical challenges involved. The heterogeneity of living tissues, from intracellular to systemic processes, complicates energy absorption and distribution.

In contrast, mEHT departs from conventional heating methods by focusing its thermal effects on cancer cell membranes and ion channels [37]. It combines thermal and nonthermal impacts, similar to the synergy between conventional hyperthermia and ionizing radiotherapy. The thermal effects provide a general background for optimizing the nonthermal processes. The thermal background determines, in general, the chemical reaction rates according to Arrhenius law The chemical reactions caused by nonionizing (bioelectric fields), ionizing (radiotherapy), and chemical changes (chemotherapy) all change their rate and efficacy by the thermal conditions (Figure 3). This approach leverages the heterogeneity of tissue thermal and electrical properties to selectively target and eliminate malignancies [37].

Cancer cells exhibit distinct temperature characteristics compared to healthy cells due to their altered metabolism and blood supply. Cancer cells generally have a higher temperature than normal cells. This is primarily attributed to their increased metabolic activity. While generally warmer, a tumor’s temperature can vary depending on factors like tumor type, size, location, and blood supply. Tumors with poor blood supply may exhibit lower temperatures in certain areas, while well-vascularized tumors tend to have more uniform temperature distribution. Cancer cells can be more sensitive to temperature changes than normal cells. Some cancer cells are more susceptible to heat-induced damage, making hyperthermia a potential treatment option for certain cancers. Cancer cells have a much higher density of chaperone proteins (mostly HSPs) than healthy cells, so by extra heat, they develop relatively fewer extra chaperones. In contrast, healthy cells massively increase their protective chaperone content. This reaction difference is one of the factors of the higher sensitivity of cancer cells to additional heat. Conversely, some cancer cells can adapt to temperature fluctuations, which may contribute to their survival and resistance to heat treatment. Understanding the temperature characteristics of cancer cells is crucial for developing and optimizing cancer therapies. Hyperthermia aims to selectively heat and destroy cancer cells while minimizing damage to healthy tissues. The temperature within the tumor microenvironment can influence various aspects of cancer progression, including cell growth, survival, angiogenesis, and immune response. While temperature can serve as a therapeutic target, it is crucial to consider the complex interplay between temperature and other factors in cancer development and treatment. Understanding these complex interactions is essential for developing more effective cancer treatments.

The nonthermal effects of mEHT involve altering ion channel activity and intracellular ion concentrations, which can impact cellular processes such as signal transduction, enzyme activity, and pH levels. The primary nonthermal processes make a broad range of interactions, which could be modified with the applied electromagnetic techniques of mEHT.

A significant process is that electromagnetic fields can nonthermally influence the activity of ion channels and transporters in cancer cells. Changes in ion channel function can alter the intracellular concentrations of ions such as calcium, sodium, and potassium. This affects various cellular processes, including signal transduction, enzyme activity, and cellular pH, which can indirectly modify the cell’s biochemical environment. The main deviation of mEHT from conventional heating techniques is that it helps and does not impede natural homeostatic actions. Constrained heating may destroy the tumor but can also cause a large attack on homeostatic regulation, which tries to compensate for the intensive local changes. So, homeostasis is loaded with two obstructions: cancer and aggressive heating (Figure 4a). The thermal and nonthermal synergy, which is optimized with a modulated RF signal in the mEHT method, helps the homeostatic regulation, induces immunogenic effects, and so extends the local effects to the whole body (Figure 4b).

mEHT alters the membrane potential of cancer cells and can affect cellular excitability and the activity of ion pumps and channels, leading to changes in intracellular ionic balance and affecting metabolic processes. Exposure to electromagnetic fields of mEHT can influence metabolic pathways in cancer cells. The mEHT-induced electromagnetic fields might impact glycolysis, oxidative phosphorylation, or lipid metabolism. These changes can influence metabolic pathways, potentially shifting the production of metabolic byproducts like lactate or reactive oxygen species (ROS). ROS can damage cellular components and contribute to tumor progression or therapy sensitivity.

mEHT also affects the ECM, altering the biochemical environment of malignant tissues. ECM composition and structure changes can influence cell adhesion, migration, and signaling pathways, potentially modifying the tumor microenvironment. Additionally, the electric field of mEHT can impact extracellular pH, affecting tumor enzyme activities and biochemical reactions. Enhanced cellular permeability due to the electric field can alter the transport of chemicals and metabolites, influencing intracellular concentrations and biochemical reactions. It can also affect protein conformation and function, impacting enzyme activity, protein interactions, and cellular signaling. Such alterations can lead to changes in metabolic pathways and cellular responses.

Notably, mEHT’s electromagnetic fields may influence gene expression by affecting transcription factors or epigenetic modifications [38]. This can alter protein synthesis related to metabolism, signaling, and cell growth, impacting overall cellular chemistry.

mEHT targets the membrane rafts of malignant cells [39,40]. It heats the transmembrane proteins of these cells similarly to nanoparticle heating [19], but without the need for artificial nanoparticles [41]. This process utilizes the differences in thermal and electromagnetic properties between malignant and healthy cells, taking advantage of the higher electric conductivity and disordered tumor microenvironment (TME) in malignant cells compared to healthy ones.

mEHT uniquely employs modulation of the radiofrequency (RF) signal. This low-frequency amplitude modulation promotes the polarization and excitation of transmembrane proteins, driving chemical reactions to induce an apoptotic process and trigger intracellular signaling pathways for immunogenic cell death. This modulation addresses the challenge of combining high-frequency RF for nano-selection with low-frequency modulation for molecular polarization. The precise targeting is based on the thermal and electric heterogeneity of living tissues, focusing on the specific characteristics of cancer cells.

Various signal pathways ensure the apoptotic processes. The external RF excites the TRAIL-FASD-FAS complex, starting an external trigger of the caspase 8 → caspase 3 → apoptosis [42] molecular excitation line. Involving mitochondria by heat, an electric field, and Bid protein, an intrinsic apoptotic pathway is formed through Bax → Cytochrome C → Caspase 9 → Caspase 3 → apoptosis [42] and a caspase-independent path, mitochondria → AIF → apoptosis [43]. The inhibitory, antiapoptotic XIAP protein is blocked by Septin 4 [44] and SMAC/Diabolo proteins [45].

### 3.5. Immune Effects of mEHT

Cancer therapies aim to destroy cancer cells while restoring healthy conditions. Cell death is possible, like apoptosis, necrosis, autophagy, necroptosis, ferroptosis, and pyroptosis. Inflammatory cell death involves types like pyroptosis and necroptosis, which are programmed forms of cell death that activate inflammatory pathways. Uncontrolled necrosis can also cause inflammation due to the chaotic release of cell contents. In contrast, apoptosis is a non-inflammatory, controlled process where cells die in an orderly manner and are removed by phagocytes without causing inflammation. Non-inflammatory cell death maintains tissue homeostasis and minimizes collateral damage.

Immunogenic cell death (ICD) is a form of cell death that activates the immune system. It is characterized by releasing damage-associated molecular patterns (DAMPs) that alert and activate immune responses. mEHT induces ICD by producing DAMPs [46], which trigger immune responses in an organized manner through harmonized thermal and nonthermal effects. These DAMPs provide information about the malignancy [47], enabling antigen-presenting cells to produce tumor-specific killer and helper T cells that recognize and destroy tumor cells through natural immune surveillance. The subsequent steps of the process are summarized in Figure 5. 

Preclinical research is a multi-step process. It begins with the concept that mEHT’s heterogeneous energy absorption enhances antitumor effects by inducing the exhaustion of heat-shock proteins (HSPs) in cancer cells [48]. Released HSPs and other DAMPs, which are designed in proper order in space and time [47], facilitate antigen presentation for tumor-specific T CD8+ cells [49], boosting the immune response against the tumor. This prepares the immune system to recognize malignant cells throughout the body, enabling systemic antitumor immune activity and potentially destroying distant micro- and macro-metastases (abscopal effect) [50]. Memory T cells retain information about the tumor, making subsequent tumor recurrence unlikely, akin to a vaccination against the cancer. This vaccination approach is patented [51].

The theoretical consideration and concept design are followed by in silico research and phantom measurements to verify thermal effects. In vitro experiments are conducted next, although they lack the complexity of living organisms. The research progresses to in vivo studies, providing feedback and refining the concept until it demonstrates sufficient accuracy, safety, and efficacy for clinical trials [52]. Preclinical studies have validated the selection process [53], confirmed immunogenic effects [54], and demonstrated the development of DAMPs [55] and ICD [49] and the abscopal effect through immune activation of killer T-cells [56]. The comparison shows the superiority of mEHT over conventional hyperthermia [42,57,58].

The addition of nonthermal effects to conventional thermal processes has been validated through several experimental and preclinical studies [42,52,59]. Phantom experiments verified thermal effects and selective heating [60]. Various laboratories measured the apoptosis and the synergetic combination of the thermal and nonthermal effects [61,62]. The experimental conclusion was that mEHT produces massive apoptosis. An overall review of the preclinical verification in vitro and in vivo [63] and immunogenic proofs [64] were published elsewhere.

Clinical efficacy validation was provided for different tumors [65,66]. See Table 1. As clinically reviewed in detail elsewhere [65,66], it was proven how clinical studies validate mEHT’s efficacy. A Phase III study showed a significant increase in survival time for patients with advanced cervical cancer [67], with observed abscopal effects [68] and improved quality of life [63]. Phase II trials have reported significant improvements with mEHT for severe cancers [66], including the frequent and hard-to-be-successful treatments for localizations such as pancreas [64,69], glioblastoma [70,71,72], non-small-cell lung cancer [73], small-cell lung cancer [74], rectal cancer [75], pain of bone metastases [76], and pediatric DIPG [77]. Additional studies and case reports support the effectiveness of combining mEHT with immune checkpoint inhibitors [78,79] and supportive therapies [80].

## 4. Conclusions

Depending on electromagnetic exposure’s strength, frequency, and duration, bioelectromagnetic interactions can influence molecular reactions. mEHT, in particular, offers a promising avenue for cancer treatment by selectively targeting malignant cells through nonthermal electromagnetic actions. These interactions can open and close membrane ion channels, alter membrane potentials, and impact cellular processes such as signal transduction and protein function, potentially influencing gene expression and chemical reactions at the molecular level.

While these effects are supported by theoretical and experimental studies, as well as preclinical and clinical validation, the full extent of bioelectromagnetic interactions on molecular reactions in living organisms remains an active research area. The potential of mEHT to extend the effects of local treatment to systemic levels is a particularly exciting area of exploration. With its superior effectiveness, ease of use, and high safety record, mEHT represents a promising advancement in oncotherapy, warranting further investigation.

## Figures and Tables

**Figure 1 curroncol-32-00158-f001:**
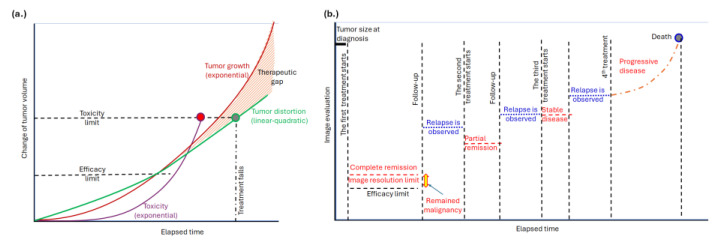
Tumor development. (**a**) There is a therapeutic gap due to tumors’ exponential growth. The toxicity limits the applicable dose. Here, the task of hyperthermia is to extend the limit with extra tumor distortion. (**b**) The available imaging observes complete remission and later relapses. The subsequent relapses happen due to the exceptional adaptation capability of the tumor cells, which remained in the location under the resolution of the actual imaging. The task of hyperthermia is to introduce new stress and suppress the possibility of adaptation.

**Figure 2 curroncol-32-00158-f002:**
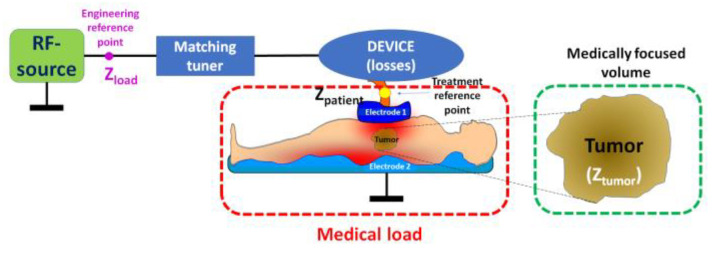
The electric circuit of modulated electro-hyperthermia. The specialized electronics regulate two reference points: the engineering control to ensure the optimal work and minimal loss of the RF source, having super-low reflected imaginary power, and the medical reference point to ensure the maximal energy focused on the selected target, the tumor. Proper material selection and careful design of specialized electronics minimize all losses.

**Figure 3 curroncol-32-00158-f003:**
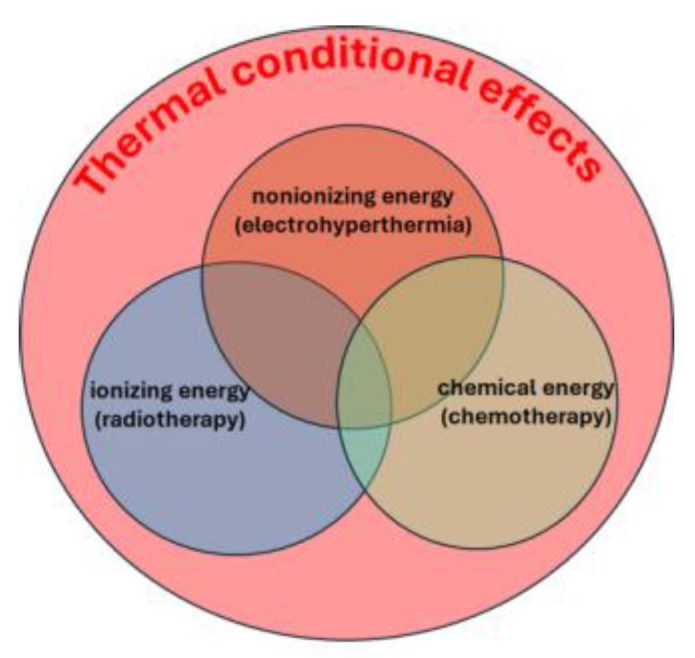
The thermal background guarantees the optimization of the nonthermal processes.

**Figure 4 curroncol-32-00158-f004:**
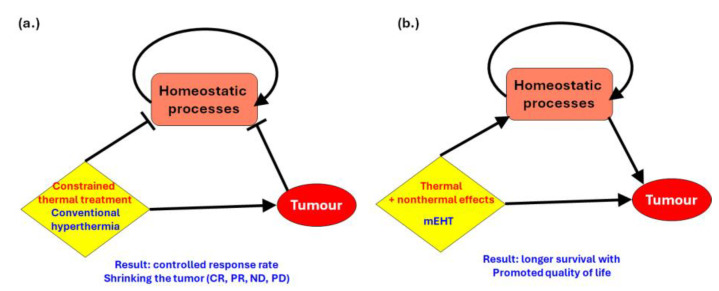
The difference between conventional hyperthermia and mEHT concerning natural homeostasis. (**a**) The effect of traditional hyperthermia. It works with thermal cell killing and has a reasonable remission rate. (**b**) mEHT works in harmony with homeostatic processes and elongates survival time with an increase in quality of life.

**Figure 5 curroncol-32-00158-f005:**
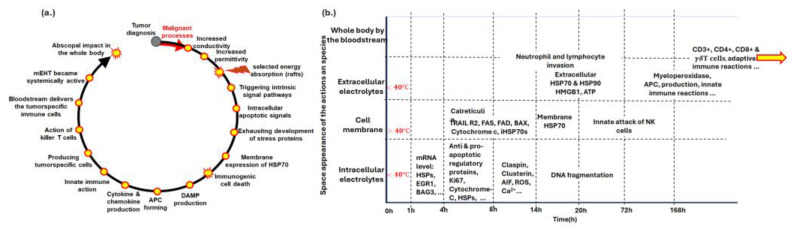
The immunogenic processes ignited by mEHT. (**a**) The steps of immunogenic development after mEHT treatment. (**b**) The molecular effects with elapsed time. The yellow arrow is intended to show that the immunogenic process continues even after 168 h.

**Table 1 curroncol-32-00158-t001:** A selection of the clinical studies using the mEHT method.

No.	Tumor Site	No. Pts.	Treatment Used	Results	Ref.
1	Advanced relapsed gliomas	12	mEHT + RT + ChT	RR = 25%. PFS = 10 m. OS = 9 m, 1yS = 25%.	Fiorentini et al., 2006 [81]
2	Mostly advanced brain gliomas	140	mEHT + RT + ChT	OS = 20.4 m. QoL: safe and well tolerated.	Sahinbas et al., 2007 [82]
3	High-grade gliomas	179	mEHT + RT + ChT	KS > 50%, AST: No. = 53, 1yS = 100%, 5yS = 51%, OS = 38.2 m, events: 19.8%, GBM: No. = 126, 1yS = 92%, 5yS = 38%, OS = 20.3 m, events: 26.4%, QoL: improved.	Hager et al., 2008 [70]
4	Relapsed gliomas	149	mEHT + RT + ChT (BSC)	Double arm, palliation setup. AST: No. = 38, mEHT No. = 28, RR: mEHT = 48%, Ctrl = 10%, 5yS: mEHT = 83%, Ctrl = 25%, OS: mEHT = 16.5 m, GBM: No. = 111, mEHT No. = 24, RR: mEHT = 29%, Ctrl = 4%, 5yS: mEHT = 3.5%, Ctrl = 1.2%, OS: mEHT = 14 m. QoL: improved.	Fiorentini et al., 2018 [71]
5	High-grade gliomas	20	mEHT + RT	PFS: 1.5–3 m, QoL: No severe side effects, well tolerated.	Solodkiy et al., 2021 [83]
6	Relapsed gliomas	164	mEHT + RT + ChT	Double arm. RR: mEHT = 41.4%, Ctrl = 33.4%, OS: mEHT = 15 m, Ctrl = 12 m. 1yS: mEHT = 55%, Ctrl = 15%, QoL: improved.	Fiorentini et al., 2020 [84]
7	Recurrent gliomas	76	ChT + mEHT	Comparison to ddTMZ historical ctrl. No.: mEHT + ddTMZ = 58 mEHT = 18, Ctrl = 79. RR(incl.SD): mEHT = 55%, mEHT + ddTMZ = 53%, events: mEHT = 67%, mEHT + ddTMZ = 67%, OS: mEHT = 14.8 m, mEHT + ddTMZ = 20.8 m, 1yS: mEHT = 22.6%, mEHT + ddTMZ = 29.5%, 5yS: mEHT = 0%, mEHT + ddTMZ = 13.5%, cost-effectiveness analysis: ddTMZ + mEHT is cost-effective.	Roussakow, 2017 [85]
8	Advanced GBM	60	mEHT + RT + ChT	PFS = 13 m. Follow-up = 17 m, OS ≈ 30 m. QoL: no added toxicity.	Van Gool et al., 2018 [86]
9	Ovarian cancer	12	ChT ± mEHT	Double combined arms: PLX, CIS. PFS: mEHT + PLX = 3 m, mEHT + CIS = 6.8 m, OS: mEHT + PLX = 11.5 m, mEHT + CIS ≈ 20 m. DLT = not observed, mEHT is feasible, safe.	Kim et al., 2021 [87]
10	Recurrent, progressive ovarian cancer	19	mEHT	PFS = 4 m, OS = 8 m, events = 18. DLT = not observed. mEHT is feasible, safe.	Yoo et al., 2019 [88]
11	Advanced cervical cancer	95	CCRT ± mEHT	No.: CCRT = 53, mEHT + CCRT = 42, NED: CCRT = 68%, mEHT + CCRT = 83%. 5yDFS: CCRT ≈ 75%, CCRT + mEHT ≈ 80%, 5yS: CCRT ≈ 81%, CCRT + mEHT ≈ 79%, mets.5yDFS: CCRT ≈ 84%, CCRT + mEHT ≈ 62%, 5yOS: CCRT ≈ 73%, CCRT + mEHT ≈ 78%.	Lee et al., 2023 [89]
12	Advanced cervical cancer ±HIV	202	CCRT ± mEHT	Prospective, double arm. No.: CCRT = 101, mEHT + CCRT = 101. Alive at 6 m after treatment: CCRT = 82.2%, CCRT + mEHT = 87.1%, LDC(RR): CCRT = 24.1%, CCRT + mEHT = 45.5%, LSFS: CCRT = 19.8%, CCRT + mEHT = 38.6%. 5yS: CCRT26%, CCRT + mEHT = 33%, DFS: CCRT = 14%, CCRT + mEHT = 32%. No significant differences in acute adverse events or quality of life between the groups.	Minnaar et al., 2022 [67]
13	Advanced cervical cancer, ±HIV	100	Phase III (RT + ChT ± mEHT	No. = 100 (preliminary data). A positive trend in survival and local disease control by mEHT. QoL: no significant differences between the groups.	Minnaar et al., 2016 [90]
14	Advanced cervical cancer	72	mEHT + RT + ChT	RR = 73.5%; SD = 14.7%. Adding mEHT increased the QoL and OS compared to historical data.	Pesti et al., 2013 [91]
15	Advanced cervical carcinoma	20	mEHT + RT + ChT	mEHT increases the peri-tumor temperature and blood flow in human cervical tumors, promoting radiotherapy + chemotherapy.	Lee et al., 2018 [92]
16	Advanced cervical carcinoma	38	mEHT + ChT	Double arm. No. ChT = 20, ChT + mEHT = 18. RR: ChT = 40%, ChT + mEHT = 72.2%. 2yS: ChT ≈ 50%, ChT + mEHT ≈ 90%.	Lee et al., 2017 [93]
17	Advanced cervical carcinoma, ±HIV	206	Phase III (CCRT ± mEHT)	Abscopal effect: CCRT = 5.56%, CCRT + mEHT = 24.07%, measured with complete metabolic response by PET of all sites of disease.	Minnaar et al., 2020 [68]
18	Advanced cervical carcinoma, ±HIV	206	Phase III (CCRT ± mEHT) [toxicity and quality of life]	mEHT addition does not increase the toxicity, QoL: improvement in social, emotional, and physical function in CR significantly higher with mEHT addition (*p* < 0.02).	Minnaar et al., 2020 [63]
19	Advanced cervical carcinoma	202	mEHT + RT + ChT	Six-month local disease-free survival (LDFS) = 38.6% for mEHT and LDFS = 19.8% without mEHT (*p* = 0.003). Local disease control (LDC) = 45.5% with mEHT, LDC = 24.1% without mEHT; (*p* = 0.003).	Minnaar et al., 2019 [94]
20	Advanced, recurrent, or metastatic breast cancer	10	mEHT + ChT/RT.	No. = 10 (all ER-positive, 1 HER2), ORR = 55%.	Nagata et al., 2021 [95]
21	Advanced NSCLC	97	mEHT + RT + Vit.C	OS: RT + Vit.C = 5.6 m, mEHT + RT + Vit.C. = 9.4 m, PFS: mEHT + RT + Vit.C. = 3 m, RT + Vit.C. = 1.85 m (*p* < 0.0001).	Ou et al., 2020 [80]
22	NSCLC	311	mEHT + RT + ChT	No. RT + ChT = 53 (historical), RT + ChT + mEHT = 258. OS: RT + ChT = 14 m, RT + ChT + mEHT = 15.8 m. Advanced subgroup: No. RT + ChT = 43 (historical), RT + ChT + mEHT = 140. OS: RT + ChT = 11 m, RT + ChT + mEHT = 14.7 m.	Szasz, 2014 [96]
23	Advanced NSCLC adenocarcinoma	4	mEHT + ChT	Survived for more than 2 years with combined therapy. No complete remission was achieved.	Lee et al., 2015 [97]
24	Advanced hepatocellular carcinoma	21	mEHT + Sorafenib	RR = 55% (no CR was observed), 6 mPFS = 38%, PFS = 5.2 m, OS = 10.4 m.	Gadaleta-Gadaleta et al., 2014 [98]
25	Advanced rectal cancer	76	mEHT + RT + ChT	RR = 33.3%, Gr.I. thermal toxicity = 26.7%.	You et al., 2020 [99]
26	Rectal cancer	120	mEHT + RT + OP	No.: RT + OP: 58, RT + OP + mEHT = 62. Downstaging: RT + OP: 67.2%, RT + OP + mEHT = 80.7%. Gastrotoxicity: RT + OP: 87.9%, RT + OP + mEHT = 64.5%, 2yDFS: RT + OP: 79%, RT + OP + mEHT = 96%.	Kim et al., 2021 [75],
27	Pancreas	73	ChT + mEHT	During the initiation of mEHT, immune markers stabilize with the treatment, but progressive disease erodes this positive effect over time. Long-time follow-up shows a significant increase in the WBC, neutrophil, and granulocyte counts, and an increase in CRP, NLR, and GLR was observed.	Dobos et al., 2024 [100]
28	Pancreas	217	ChT ± mEHT	No.: ChT = 128, ChT + mEHT = 89. OS: ChT = 9 m, ChT + mEHT = 20 m. RR: ChT = 24%, ChT + mEHT = 45%. PD: ChT = 31 ChT + mEHT = 4.	Fiorentini et al., 2023 [69]
29	Pancreas	158	ChT + mEHT	No.: ChT = 100, ChT + mEHT = 58. OS: ChT = 11.02 m, ChT + mEHT = 19.5 m. RR: ChT = 58%, ChT + mEHT = 95%. Toxicity did not differ in the two groups, only 8 pts. had Gr.1–2 burns.	Fiorentini et al., 2021 [64]
30	Pancreas ductal adenocarcinoma	78	mEHT + ChT	Non-operable patients. No.: ChT = 39, ChT + mEHT = 39. OS: ChT = 14.19 m, ChT + mEHT = 16.96, PFS: ChT = 8.53 m, ChT + mEHT = 11.87, 1yS: ChT = 16%, ChT + mEHT = 26%. 2yS: ChT = 5%, ChT + mEHT = 6%, 1yPFS: ChT = 7%, ChT + mEHT = 11%.	Petenyi et al., 2021 [101]
31	Advanced pancreas carcinoma	106	mEHT + RT + ChT	After 3 m, PR = 22 (64.7%), SD = 10 (29.4%), PD = 2 (8.3%) with mEHT after 3 m of the therapy. In the group without mEHT at the same time, PR = 3 (8.3%), SD = 10 (27.8%), PD = 23 (34.3%). The median OS = 18 m with mEHT and OS = 10.9 m without mEHT.	Fiorentini et al., 2019 [102],
32	Advanced pancreas carcinoma	133	mEHT + RT + ChT	Two centers. No.: PFY = 26, HTT = 73, Ctrl. = 34. Conventional therapies failed. Distant metastases: PFY = 59%, HTT = 88%. OS: PFY = 12 m, HTT = 12.7 m, Ctrl = 6.5 m. 1yS: PFY = 46.2%, HTT = 52.1%, Ctrl. = 26.5%. QoL was improved.	Dani et al., 2008 [103]
33	Metastatic pancreas carcinoma	26	mEHT + ChT	RR = 71%, PFS = 3.9 m. OS = 8.9 m.	Volovat et al., 2013 [104]
34	Gastrointestinal cancer	49	mEHT + CCRT + IMT	OS: CRC and GC ≈ 20 m CCC and GC ≈ 10 m. The IMT + mEHT had the most positive effect on overall survival (HR: 0.3055; *p* = 0.0260), IL-2 and low-dose ipilimumab showed a positive tendency.	Kleef et al., 2023 [105]
35	Advanced gastric cancer	24	mEHT + MSCT + HBOT	CR = 88.0%, follow-up = 23.9 m. OS = 39.5 m, PFS = 36.5 m.	Iyikesici et al., 2020 [106]
36	Advanced abdominal soft-tissue sarcoma	24	mEHT + ChT	RR = PR + SD = 88%	Volovat et al., 2014 [107]
37	Various sites	30	RT + mEHT	CR = 22.2%, PR = 55.5%, SD = 14.8%, PD = 7.4%.	Chi, 2020 [108]
37	Various sites	131	IMT + mEHT	RR = 31.3%, PFS = 10 m, 1yS = 66.5%, 2yS = 36.6%.	Kleef et al., 2020 [79]
39	Various sites (advanced, metastatic)	784	mEHT + RT + ChT + OP	GBM (n = 36, OS = 2.2y); breast cancer (n = 72, OS = 5y), coloncancer (n = 79, OS = 4y) NSCLC (n = 54, OS = 1.5y), PC (n = 27, OS = 1.1y), soft-tissue sarcoma (n = 16, OS = 3.1y), melanoma (n = 12, OS = 2.6y); ovarian cancer (n = 33, OS = 3.5y), kidney cancer (n = 13, OS = 4.1y), prostate cancer (n = 25, OS = 8.3y), rectal cancer (n = 13, OS = 4.0y), uterine cancer (n = 13, OS = 4.0y), head and neck cancer (n = 13, OS = 4.0y), lymphoma (Hodgkin and non-Hodgkin) (n = 13, OS = 4.0y), cholangiocarcinoma (n = 13, OS = 4.0y).	Parmar et al., 2020 [109]
40	Primary, recurrent, and metastatic sarcomas	13	mEHT + RT + ChT	Cases. Good response.	Jeung et al., 2015 [110]
41	Pelvic and spinal bone metastases	61	RT + mEHT	Pain intensity: before = 37, after = 13. Frequency of occurrence of BPT: before = 33, after = 13. Relative frequency taking painkillers: before = 66, after = 62. Duration of pain release 1.5 weeks longer with RT + mEHT than with RT of mEHT alone.	Kim et al., 2024 [76]
42	Peritoneal carc. with malignant ascites	260	mEHT + TCM	No.: mEHT + TCM = 130, TCM = 130. RR: mEHT + TCM = 77.69%, TCM = 63.8 (*p* = 0.005). KPS: mEHT + TCM = 49.23%, TCM = 32.3% (*p* < 0.05).	Pang et al., 2017 [111]
43	Metastatic cancers (colorectal, ovarian, breast)	23	mEHT + ChT	RR: 85.7% (80 days), 72.2% (160 days). OS (mean) = 497 days, PFS (mean) = 339 days.	Ranieri et al., 2017 [112]
44	Metastatic/recurrent cancers (different types)	33	mEHT + RT	RR = 87.9%. Autoimmune toxicity = 9.1% (abscopal effects were observed).	Chi et al., 2020 [78]

CCRT = concurrent chemoradiotherapy, DFS = median disease-free survival, PFS = median progression-free survival, OS = median overall survival, RR = response rate, CR = complete remission, LDC = local disease control, LDFS = local disease-free survival, 5yS = five-year survival, 1yS = one-year survival, Crtl. = control, ddTMZ = dose-dense temozolomide, CCC = cholangiocellular cancer, GC = gastric cancer, EC = esophageal cancer, CRC = colorectal cancer, PC = pancreatic cancer, MSCT = metabolically supported chemotherapy, HBOT = hyperbaric oxygen therapy, BTP = breakthrough pain, TCM = traditional Chinese medicine, KPS = Karnofsky performance score.

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
