# Peer review of "Bioelectromagnetism for Cancer Treatment—Modulated Electro-Hyperthermia"

_curroncol, 2025, doi:10.3390/curroncol32030158_

Round 1

Reviewer 1 Report

Comments and Suggestions for Authors

I have read this manuscript with interest. It is quite well written, and the Introduction invites one to pursue deeper reading. However, it suffers from a main drawback: it is an interesting story, but where results, details and methodology are absent. Without that, it appears as a reference search. This is an unfair feeling, I understand, and hence I suggest the authors to complete the present study. These are my specific comments:

1.       Line 138: Warburg effect description repeated

2.       Figure 2 is just a cartoon. More details are needed at this point. How is the technique really implemented?

3.       Line 159: please extend a little your description of the relaxations

4.       Section 3.4: results beyond the scheme of Fig 3 are needed

5.       In general, detailed results, even if just reproduction of already published materials are missing

Author Response

Dear Reviewer,

please see my point-by-point reply in the attachment.

Kind regards,

Andras Szasz

Reviewer 2 Report

Comments and Suggestions for Authors

Title: Bioelectromagnetism for cancer treatment – the modulated electro-hyperthermia

Author: Andras Szasz

General Comments:

o    This review explores the potential of bioelectromagnetism to br used as oncological treatments. Nowadays, the number of the cancerous metastases increase exponentially, so the development of efficient treatments has become crucial. Thus, the subject of this paper is of interest for the oncotherapy.

o   The structure of the paper fulfills the structure of a research article.

o   6 keywords are included by the authors.

  • The Introduction section provide sufficient background information for readers in the immediate field to understand the problem that this study addresses.

Despite the fact that the review is interesting, before publication, the next issues should be clarified:

1.   Please add a section describing the temperature characteristics of cancerous cells;

2.      Please define the Warburg effect at page 3 line 83 (where it first appears in the text);

3.      Page 4, line 127: please delete the mRHT abbreviation (it was defined in line 125);

4.      Page 4, line 145: please explain what “pure electric field” means;

5.      Page 8, figure 5a: please increase the font.

6.      If I counted correctly and it is not a coincidence of names, 35% of the references belong to the author and in my opinion this percentage should be reduced

Author Response

(The authors gave the same response as above.)
